# Investigating Enhanced Microwave Absorption of CNTs@Nd_0.15_-BaM/PE Plate via Low-Temperature Sintering and High-Energy Ball Milling

**DOI:** 10.3390/ma17143433

**Published:** 2024-07-11

**Authors:** Chengying Wang, Xiaohua Feng, Chengwu Yu, Lixia Zhang, Shengguo Zhou, Yi Liu, Jing Huang, Hua Li

**Affiliations:** 1School of Materials Science and Engineering, Jiangxi University of Science and Technology, Ganzhou 341000, China; 2Zhejiang-Japan Joint Laboratory for Antibacterial and Antifouling Technology, Ningbo Institute of Materials Technology and Engineering, Chinese Academy of Sciences, Ningbo 315201, China; 3Zhejiang Engineering Research Center for Biomedical Materials, Institute of Biomedical Engineering, Ningbo Institute of Materials Technology and Engineering, Chinese Academy of Sciences, Ningbo 315201, China; 4Beijing Institute of Control Engineering, Beijing 100190, China

**Keywords:** barium ferrite, CNTs, PE, neodymium-doped, microwave absorption

## Abstract

Composite plates comprising a blend of rare earth neodymium-(Nd) doped M-type barium ferrite (BaM) with CNTs (carbon nanotubes) and polyethylene WERE synthesized through a self-propagating reaction and hot-pressing treatment. The plates’ microscopic characteristics were analyzed utilizing X-ray diffraction (XRD), Fourier transform infrared spectrophotometry (FTIR), thermo–gravimetric analysis (TGA), Raman, and scanning electron microscopy (SEM) analytical techniques. Their microwave absorption performance within the frequency range of 8.2 to 18 GHz was assessed using a vector network analyzer. It showed that CNTs formed a conductive network on the surface of the Nd-BaM absorber, significantly enhancing absorption performance and widening the absorption bandwidth. Furthermore, dielectric polarization relaxation was investigated using the Debye theory, analyzing the Cole–Cole semicircle. It was observed that the sample exhibiting the best absorbing performance displayed the most semicircles, indicating that the dielectric polarization relaxation phenomenon can increase the dielectric relaxation loss of the sample. These findings provide valuable data support for the lightweight preparation of BaM-based absorbing materials.

## 1. Introduction

The escalating electromagnetic pollution and the pressing need for stealth technology advancement have severely restricted the practical utility of conventional absorbing materials [1]. Their drawbacks, such as limited absorption spectra, high density, and insufficient thermal stability, have impeded their widespread adoption [2,3,4]. Consequently, there has been a surge in the quest for lightweight and efficacious microwave attenuation materials emerging as a focal area of research [5,6,7]. Concurrently, there is a growing emphasis on enhancing electron polarization and refining impedance matching in absorbing materials, garnering increased attention from international scholars [8,9]. For instance, the synthesis of wide-bandwidth, high-absorption absorbers have been achieved by combining non-destructive or low-consumption dielectric materials with high-efficiency absorbers.

Commonly utilized absorption materials encompass carbon nanotubes (CNTs), MXene, ceramics, and various ferrites (spinel, magnetite, garnet), with ferrite particularly distinguishing itself among traditional options due to its notable absorption efficiency, thin coating, and wide bandwidth capabilities [10,11]. Furthermore, owing to variations in O^2−^ repetitions and the interval between occurrences of the Ba^2+^ ion layer, magnetite ferrite showcases diverse structures, rendering it suitable for use as both hard magnetic materials and very high-frequency soft magnetic materials, thus finding extensive applications. M-type barium ferrite (BaM) serves as a prototypical simple magnetite ferrite renowned for its high resistivity, excellent chemical stability, and remarkable magnetic properties, characterized by a hexagonal symmetrical structure [12]. Nevertheless, the practical utilization of BaM is considerably restricted by its drawbacks, including high density and narrow absorption band. To overcome these limitations, researchers frequently employ methods such as ion doping and recombination for modification.

The table reveals that ion doping and composite materials with carbon can effectively enhance the absorption performance of BaM (Table 1), where RL denotes the reflection loss of the sample, and RL_min_ denotes the minimum reflection loss value of the sample. Ionic doping primarily involves rare earth doping and excessive metal doping.

Rare earth ion doping initially boosts the magneto–crystalline anisotropy field of the material, thereby augmenting magnetic loss. Subsequently, the alteration of grain size reduces resistivity, amplifying the annular current induced by incident electromagnetic waves within the material, thereby enhancing eddy current loss. Lastly, the disparate radii of metal ions in rare earth ions and BaM lead to lattice distortion, thereby altering the lattice constant and enhancing dielectric loss.

BaM exhibits anisotropy, with its pronounced axial anisotropy, leading to a low dielectric constant and high resonance frequency, rendering it less than ideal as a standalone absorber. Consequently, researchers leverage the favorable electromagnetic wave absorption properties of carbon materials and barium ferrite to create composites that complement each other, achieving magnetoelectric synergy [27]. This approach allows for not only ion doping of BaM materials but also further compounding with carbon materials, thereby enhancing the overall absorption performance. From the above table, the composite material of Nd-doped BaM with graphene oxide (GO) exhibits superior microwave absorption performance compared to other rare-earth doped materials. When Nd^3+^ is doped into the magneto plumbite structure, it can replace Fe^3+^, causing lattice distortions. Its paramagnetic properties can improve the magnetic properties of BaM.

Therefore, we continued our previous choice, Nd-doped BaM as the absorber, with CNTs as a composite, which differs from the GO composite mentioned above. GO, rich in oxygen-containing functional groups on its surface and edges, exhibits excellent dielectric properties that enhance dielectric loss and improve wave absorption performance. In contrast, CNTs, due to their high electrical conductivity, effectively dissipate electromagnetic waves through electronic polarization. Additionally, GO’s single atomic layer structure provides extra interfaces for electromagnetic wave loss and creates multiple scattering centers, enhancing the absorption effect. Meanwhile, CNTs may form extensive conductive networks on barium ferrite particles, converting electromagnetic waves into resistive heat for dissipation. This study also involves compounding polyethylene (PE) with high-efficiency absorbing agents and using hot-pressing techniques to produce absorbers of specific shapes and thicknesses [28]. The absorption performance under different mass ratios was thoroughly investigated with the goal of constructing broadband and high-performance absorbers to expand their potential applications in wave absorption [29,30,31].

A comparative analysis of the microwave absorption performance among the modified BaM materials was investigated. The results demonstrated that the absorbing materials developed herein exhibit distinct advantages in terms of microwave absorption capabilities when compared to those reported in previous research (Figure 1).

## 2. Experiment Section

### 2.1. Raw Materials

CNTs were procured from Shenzhen Hongda Chang Evolution Co., Ltd. (Shenzhen, China) PE, ferric nitrate nonahydrate (Fe(NO_3_)_3_·9H_2_O), barium nitrate (Ba(NO_3_)_2_), neodymium nitride (Nd(NO_3_)_3_·6H_2_O), citric acid monohydrate (C_6_H_8_O_7_·H_2_O), and ammonia water (NH_3_·H_2_O) were all acquired from Sinopharm Chemical Reagent (Shanghai, China).

### 2.2. Preparation Method

The following were dissolved in 500 mL of distilled water: 143.62 g of Fe(NO_3_)_3_·9H_2_O; 7.84 g of Ba(NO_3_)_2_; 1.97 g of Nd(NO_3_)_3_·6H_2_O; and 81.95 g of C_6_H_8_O_7_·H_2_O. The mixture was stirred well with a magnetic blender for 30 min before being transferred to a three-necked flask. The pH was adjusted to approximately 7.0 with ammonia. Then, the solution was heated to 80 °C in a water bath and maintained at this temperature for about 5–6 h until the solution in the three-necked flask became viscous, forming a gel. The gel was then poured into a beaker for later use. The gel obtained above was placed into a muffle furnace with a crucible and heated from room temperature to 200 °C at a heating rate of 2 °C/min. It was kept at 200 °C for 2 h, during which the gelatinous liquid underwent a self-propagation combustion reaction, resulting in the formation of a dendritic sample. The sample was subsequently crushed in an agate mortar, and the obtained powder was subjected to further sintering in a muffle furnace. The temperature was increased from room temperature to 850 °C at a heating rate of 5 °C/min and maintained at 850 °C for 3 h to obtain Nd_0.15_-BaM powder. CNTs, with a mass ratio of 8%, were then added to the Nd_0.15_-BaM powder. Different proportions of PE were mixed with Nd_0.15_-BaM/8%CNTs (1:2, 1:1, 2:1, 3:1, 4:1), and the resulting mixture was ball milled using ball-milling equipment with iron balls and agate tanks. The mass ratio of iron balls to powder was 10:1, and the mass ratio of large balls to small balls was 7:1, where the diameter of the large ball was 10 mm, and the diameter of the small ball was 6 mm. The agate jar rotated at 300 rpm/min for 4 h. Subsequently, an appropriate amount of powder was hot-pressed at 5 MPa and 130 °C and insulated for 8 min to obtain PE/Nd_0.15_-BaM@8%CNTs bulk materials with different mass ratios.

### 2.3. Testing and Characterization

The phase composition of the sample was determined using an X-ray diffractometer (XRD, D8 Advance, Bruker, Bremen, Germany) equipped with a Cu target and Kα radiation. The instrument operated at a voltage of 40 kV and a current of 40 mA, with a step size of 0.1 nm, scanning speed of 0.1 s/step, and scanning range of 5 to 90°. The microscopic topography of the material was observed using scanning electron microscopy (SEM, Zeiss Gemini 300, Zeiss, Oberkochen, Germany). The surface composition of the sample was identified using a Fourier transform infrared spectrometer (FTIR, Nicolet IS 50, Thermo, Norristown, PA, USA).

Samples were analyzed for electromagnetic parameters in the 8.2 to 18 GHz range using a vector network analyzer (VNA, Keysight P9377B, Agilent, Santa Clara, CA, USA). Depending on the waveguide segment, the sample was cut to the appropriate size. The cut sample was then placed into the waveguide cavity, and the ε and μ of the sample were measured using the vector network analyzer waveguide method. According to the transmission line theory, the reflection loss RL (dB) and normalized characteristic impedance (dimensionless quantity) of the absorbing material were calculated using Equations (1) and (2) [32].
(1)$$RL=20log_{10}\left|\frac{{\hat{z}}_{in}-Z_{0}}{{\hat{z}}_{in}+Z_{0}}\right|$$
(2)$$Z=|{\hat{Z}}_{\mathrm{in}}/Z_{0}\left|=\right|\frac{Z_{0}\sqrt{\frac{\mu_{r}}{\varepsilon_{r}}}\tan h\left|j\left(\frac{2\pi fd}{c}\right)\sqrt{\varepsilon_{r}\mu_{r}}\right|}{Z_{0}}|$$
where $Z=\left|\frac{{\hat{Z}}_{in}}{Z_{0}}\right|$ Quantifies the magnitude of the normalized characteristic impedance, *Z_0_* is the impedance of the free space, ${\hat{Z}}_{in}$ denotes the normalized input impedance, *d* stands for the thickness of the absorbing material, *ε_r_* represents the complex permittivity of the material, *μ_r_* signifies the complex permeability of material, *f*, and *c* denote the frequency and velocity of electromagnetic waves in free space, respectively.

We calculated the *C*_0_ value of the samples using the following formula [33]:(3)$$C_{0}=\mu \prime \prime {\left(\mu \prime \right)}^{-2}f^{-1}$$

*μ″*, *μ′*, and *f* denote the imaginary part, the real part of the material’s complex permeability, and the frequency of the electromagnetic wave, respectively.

According to Debye’s theory, the relationship between ε′ (the real part of permittivity) and ε″ (the imaginary part of permittivity) can be described by the following equation:(4)$${\left(\varepsilon^{\prime }-\frac{\varepsilon_{s}-\varepsilon_{\infty }}{2}\right)}^{2}+{\left(\varepsilon^{\prime \prime }\right)}^{2}={\left(\frac{\varepsilon_{s}-\varepsilon_{\infty }}{2}\right)}^{2}$$
where *ε_s_* is the static permittivity and *ε_∞_* is the relative permittivity at infinite frequency.

## 3. Results and Discussion

From the XRD spectra, the diffraction peaks of coatings with different ratios of PE: CNTs@Nd_0.15_-BaM closely matched the characteristic peaks of BaM. However, there was a slight leftward shift in peak position compared to the characteristic peaks of BaM (Figure 1). The 2θ angles at 30.83°, 32.20°, 34.11°, 37.08°, 40.32°, 42.42°, 55.06°, and 56.33° corresponded to the (008), (107), (114), (203), (205), (206), (217), and (304) crystal planes of BaM (JCPDS No.43-0002) [34]. The slight leftward shift suggests that Nd^3+^ partially occupied the position of Fe^3+^, leading to a slight increase in the crystal plane spacing due to the larger radius of Nd^3+^ compared to Fe^3+^ (Nd^3+^ = 99.5 pm, Fe^3+^ = 64 pm) [21].

The *x*-axis of Figure 2 represents the scanning from an angle of 5–90°, and the *y*-axis represents the relative intensity of the sample’s physical phase. According to the law of Plague: *2dsinθ = nλ*, where *d* is the interplanar spacing, *λ* is the wavelength of the incident wave, *θ* is the angle between the incident light and the crystal plane, and *n* is an integer multiple of the wavelength [6]. In the absence of metallographic changes, when the larger ion partially replaced the smaller ion, the *d* value increased, resulting in a slight decrease in the *θ* value. Thus, our self-spreading growth and high-temperature roasting process successfully yielded Nd-doped BaM powder for coating preparation.

Normal CNTs exhibit diffraction peaks at approximately 26° and 44°, representing the (002) and (100) crystal planes, respectively. However, in the composite coating, the diffraction peaks at these two positions were nearly invisible (Figure 2). This is likely due to the inherently low peak intensity of CNTs at these locations, and after compounding with BaM, the peak intensity of BaM became significantly higher than that of CNTs, causing the characteristic peaks of CNTs to diminish. Consequently, after being prepared into coatings, the characteristic peaks of CNTs were almost indiscernible.

The *x*-axis of Figure 3 represents the wave number test range of 4000 cm^−1^ to 300 cm^−1^, and the *y*-axis represents the relative intensity of the sample transmittance. According to the infrared spectra, the specific vibration peaks of -CH_2_- appeared at 2913 cm^−1^, 2847 cm^−1^, 1463 cm^−1^, and 718 cm^−1^, indicating the presence of PE as a binder in the bulk absorber during the hot-pressing process. The absorption peaks at 432 cm^−1^, 533 cm^−1^, and 574 cm^−1^ corresponded to the characteristic absorption peaks of BaM (Figure 3). Peaks within the 400 to 440 cm^−1^ range represented lattice vibration peaks of octahedral metal ions, while those within the 580 to 610 cm^−1^ range represented lattice vibrational absorption peaks of tetrahedral metal ions [13]. These findings suggest that during the preparation process, the BaM powder bonded together by PE, forming blocks.

The *x*-axis of Figure 4 represents the wave number test range of 2500 cm^−1^ to 800 cm^−1^ in the Raman test, and the *y*-axis represents the relative intensity of the Raman spectra of the samples. To further verify the presence of CNTs, we performed Raman spectroscopy on the samples. The D peak, centered around 1350 cm^−1^ and indicative of lattice defects and disorder in graphitic materials, along with the G peak at approximately 1580 cm^−1^, associated with the in-plane stretching vibrations of sp^2^ hybridized carbon atoms, confirmed the ordered structure of graphitic materials and substantiated the presence of CNTs (Figure 4). These results suggest that CNTs were retained throughout the ball milling and hot-pressing processes and remained present in the sample. Additionally, the graphitization degree was reflected by the intensity ratio of the D to G peaks (I_D_/I_G_). 

It is evident that as the content of CNTs@Nd_0.15_-BaM decreased, the phenomenon of BaM enrichment diminished. At a ratio of 2:1, CNTs@Nd_0.15_-BaM was distributed in PE in a relatively uniform network form. However, with an increase in the PE content, the distribution of BaM in PE became uneven and discontinuous (Figure 5). The distribution of BaM in PE should have a great impact on the absorption performance.

From the thermogravimetric analysis (TGA) diagram of different mass ratios between PE and CNTs@Nd_0.15_-BaM, it is observed that when the temperature exceeded 200 °C, PE began to soften and gradually underwent combustion, resulting in the production of CO_2_ and water vapor, leading to a sharp decrease in sample mass. The sample mass tended to stabilize at around 500 °C.

It is notable from the curves that the PE content in all samples appeared slightly higher than the predetermined ratio (Figure 6). This discrepancy could be attributed to two possible reasons. First, all samples may have adsorbed a small amount of water molecules, resulting in less than 100% thermogravimetric loss compared to pure PE samples. Second, the composite CNTs in the samples might have undergone oxidation at 400 °C, generating CO_2_. The remaining mass would then primarily consist of Nd_0.15_-BaM ceramic samples.

The hysteresis loop illustrates the relationship between saturation magnetization and coercivity of samples with varying proportions (Figure 7). In the composites, the saturation magnetization decreased with an increasing PE content, declining from 30.67 emu/g to 9.57 emu/g. This decrease in saturation magnetization can be attributed to the increase in the non-magnetic component of PE. The hysteresis loop of PE appeared as a horizontal straight line, indicating its non-magnetic nature. However, despite the increase in PE content, the coercivity of the sample remained relatively high, consistently exceeding 7 kOe. This suggests that the absorption performance of the sample was primarily reliant on electromagnetic loss.

The electromagnetic wave-absorbing properties of PE/CNTs@Nd_0.15_-BaM with different mass ratios were examined. It is evident that the optimal absorbing ability was achieved when the PE: CNTs@Nd_0.15_-BaM ratio was 2:1 compared to other ratios (Figure 8). This is primarily attributed to the fact that electromagnetic waves can readily penetrate through the low-content CNTs@Nd_0.15_-BaM (PE: CNTs@Nd_0.15_-BaM ratio exceeding 2:1) due to the weak absorbing effect of PE. Conversely, the agglomeration of CNTs@Nd_0.15_-BaM and the discontinuous absorption channels led to poor impedance matching and inferior absorbing properties for high-content CNTs@Nd_0.15_-BaM (PE: 8%CNTs@Nd_0.15_-BaM ratio less than 2:1).

Specifically, when the mass ratio of PE and 8%CNTs@Nd_0.15_-BaM was 2:1, it exhibited *RL_min_* values of −58.01 dB with an effective absorption bandwidth (*EAB*) of 4.26 GHz at a thickness of only 1.9 mm. Considering factors such as *RL_min_* value, *EAB*, and composite content, the composite of PE and 8%CNTs@Nd_0.15_-BaM at a ratio of 2:1 demonstrated stable and outstanding absorbing performance.

When the mass ratio of PE to 8%CNTs@Nd_0.15_-BaM was 1:2 and 4:1, the samples exhibited very weak absorption performance (Figure 9). This is consistent with the fact that PE itself does not possess effective absorption properties. However, at a mass ratio of 1:1, a maximum effective absorption bandwidth of 5.68 GHz was observed at a thickness of 1.8 mm. At a mass ratio of 2:1, the maximum effective absorption bandwidth was 6.44 GHz at a thickness of 2.2 mm, with a maximum absorption peak of −40.21 dB at 13.86 GHz. Furthermore, the maximum absorption peak was −58.01 dB at 16.43 GHz at a thickness of 1.9 mm, with an effective absorption bandwidth of 4.26 GHz. At a thickness of 2.5 mm, there was a maximum effective absorption bandwidth of 4.79 GHz, while at 14.53 GHz, there was a maximum absorption bandwidth of −16.31 dB. Additionally, the maximum absorption peak occurred at a thickness of 2.9 mm, at 11.81 GHz, with a magnitude of −24.73 dB, and an effective absorption bandwidth of 3.72 GHz.

To further investigate the absorption mechanism of electromagnetic waves, we used a vector network analyzer test to obtain the relative complex permittivity and complex permeability of the composite samples in different proportions. Where the units of both the real and imaginary parts of the complex permittivity were F/m, the units of both the real and imaginary parts of the complex permeability were H/m, and the tangent of the complex permittivity and the tangent of the complex permeability were dimensionless quantities, both the real and imaginary parts of the complex permittivity decreased with the decrease in 8%CNTs@Nd_0.15_-BaM content (Figure 10). The real part of the complex permeability exhibited slight fluctuations with frequency, albeit remaining relatively low. This phenomenon can be attributed to the displacement rate and spin rate of the domain wall. The imaginary part of the complex permeability signified the loss of magnetic energy arising from the interaction of domain wall motion and spin moment in the magnetic field. However, within the 8.2–18 GHz range, the contribution of domain wall motion was deemed negligible, predominantly attributed to the influence of magnetic moment [35]. The imaginary part also exhibited slight fluctuations with frequency, yet a notable formant peak was observed at 11–12 GHz when the content of CNTs@Nd_0.15_-BaM was high at 8%. This phenomenon is attributed to the magnetic loss of Nd-doped barium ferrite, primarily stemming from natural resonance. It indicates that the addition of neodymium enhanced the magnetic loss of the composite. Furthermore, since the addition of neodymium ions augmented the net magnetic moment of barium ferrite, the real part of its complex permeability was higher at higher contents of 8%CNTs@Nd_0.15_-BaM compared to lower contents.

In the 8.2–18 GHz frequency range, the materials predominantly exhibited natural ferromagnetic resonance and eddy current loss. To further explore the magnetic loss characteristics across various ratios of PE to 8%CNTs@Nd_0.15_-BaM, we calculated the C_0_ value of the samples using Formula (3). A constant C_0_ value signified the predominance of eddy current losses. Analysis of the C_0_ values and their relationship with frequency *f* (Figure 11) revealed that the curves for all six samples remained nearly flat across the 8.2–18 GHz range. This observation strongly suggests that eddy current loss was the dominant mechanism within this frequency band.

The Cole–Cole curves of the samples all displayed a pronounced upward tail (Figure 12), which signifies transmission loss resulting from current generation [36]. 

Figure 12a–e show the relationship between ε” and ε’ for different samples. According to previous studies, the pronounced upward tail in samples (a)–(e) indicates the presence of conduction losses. It is clear that samples (a)–(f) showed 1, 2, 3, 2, 1, and 0 Cole-Cole semicircles, respectively. The PE and CNTs@Nd_0.15_-BaM, with a mass ratio of 2:1, had the largest number of Cole–Cole semicircles. The Cole–Cole curves of the PE and CNTs@Nd_0.15_-BaM composite material exhibited several distinct semicircles, which were a result of polarization relaxation processes [37]. This showed the strongest polarization loss capability. Notably, a large number of voids and defects within the polyethylene and CNTs@Nd_0.15_-BaM composites formed dipole defects that led to dipole polarization relaxation, which can effectively attenuate electromagnetic waves.

To summarize, based on the Cole–Cole semicircle curves, the 2:1 sample produced the most Cole–Cole semicircles, indicating that the polarization relaxation phenomena were most significant in sample 2:1.

Z = |Z_in_/Z_0_| quantifies the magnitude of the normalized characteristic impedance, which serves to characterize the material’s impedance-matching capabilities. Theoretically, an |Z_in_/Z_0_| value approaching 1 indicates optimal impedance matching, enabling electromagnetic waves to be fully absorbed and converted into thermal energy [38]. The sample with a 2:1 ratio exhibited the largest area under the curve in the 0.8–1.2 GHz frequency range, signifying the most effective impedance matching among the tested samples (Figure 13). Consequently, this sample demonstrated superior microwave absorption performance relative to its counterparts.

## 4. Conclusions

The present study undertook the fabrication of a series of PE/CNTs@Nd_0.15_-BaM composite sheets via the sol–gel method, ball milling process, and hot-pressing molding process. Through process optimization and performance comparison, it was discovered that when the mass ratio of PE to CNTs@Nd_0.15_-BaM was 2:1, the sample demonstrated the most promising microwave absorption performance. At a thickness of 2.2 mm, the maximum absorption reached −40.21 dB at 13.86 GHz, accompanied by an effective absorption bandwidth of 6.44 GHz. Meanwhile, at a thickness of 1.9 mm, the maximum absorption recorded was −58.01 dB at 16.43 GHz, with an effective absorption bandwidth of 4.256 GHz. By employing the Debye theory, the microwave absorption mechanism of the composite materials was scrutinized. It was observed that the sample with the optimal ratio exhibited an internal dielectric relaxation phenomenon, thereby augmenting the dielectric relaxation loss and ultimately enhancing its microwave absorption performance.

## Figures and Tables

**Figure 1 materials-17-03433-f001:**
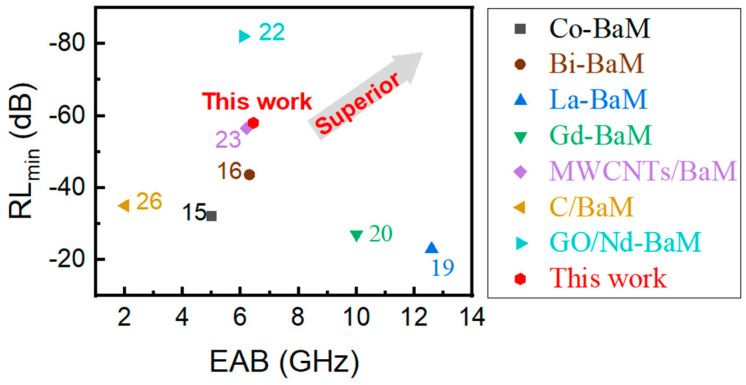
Comparison of the electromagnetic wave absorption properties of various materials: *RL_min_* and *EAB*.

**Figure 2 materials-17-03433-f002:**
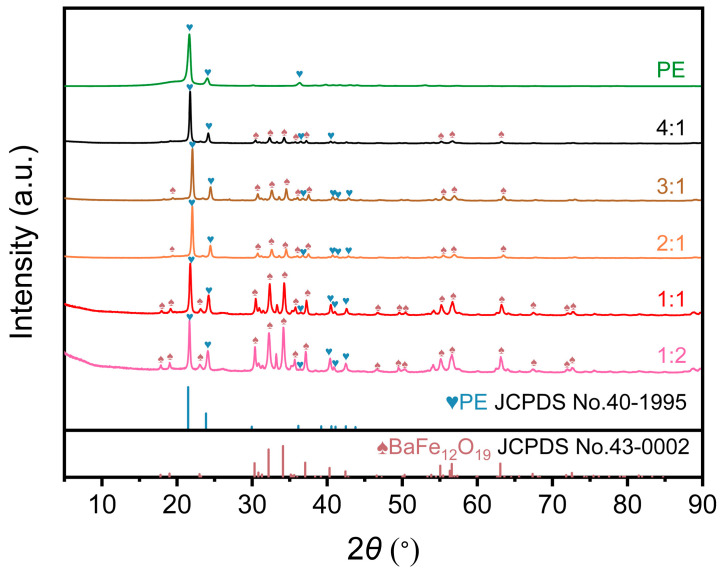
XRD image of different mass ratios between PE and CNTs@Nd_0.15_−BaM [34].

**Figure 3 materials-17-03433-f003:**
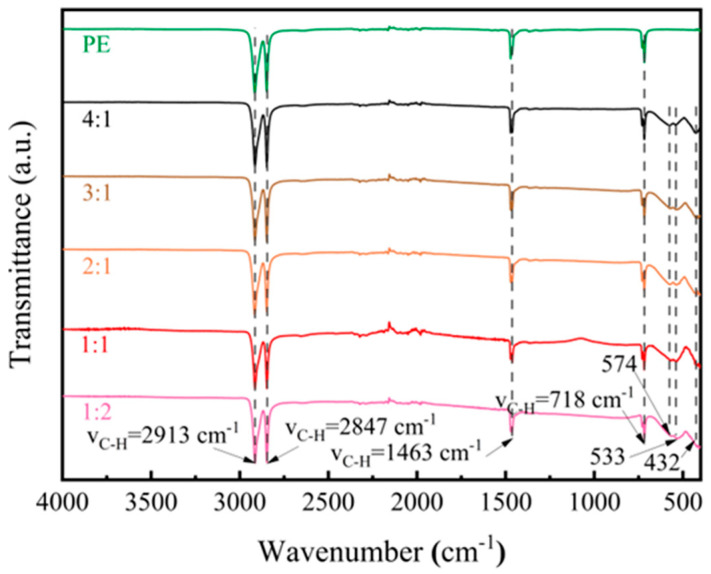
FTIR spectrogram of different mass ratios between PE and CNTs@Nd_0.15_-BaM [34].

**Figure 4 materials-17-03433-f004:**
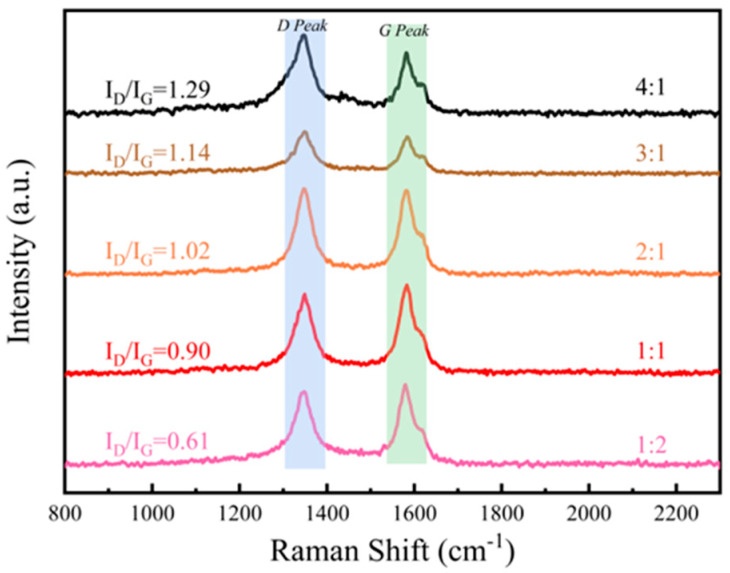
Raman spectra of different mass ratios between PE and CNTs@Nd_0.15_-BaM.

**Figure 5 materials-17-03433-f005:**
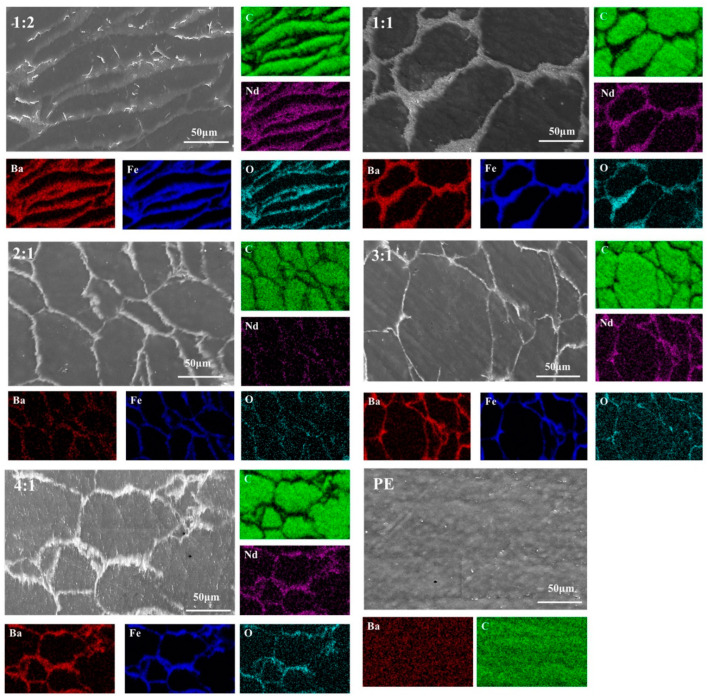
SEM images and EDS of different mass ratios between PE and CNTs@Nd_0.15_-BaM (1:2, 1:1, 2:1, 3:1, 4:1, PE).

**Figure 6 materials-17-03433-f006:**
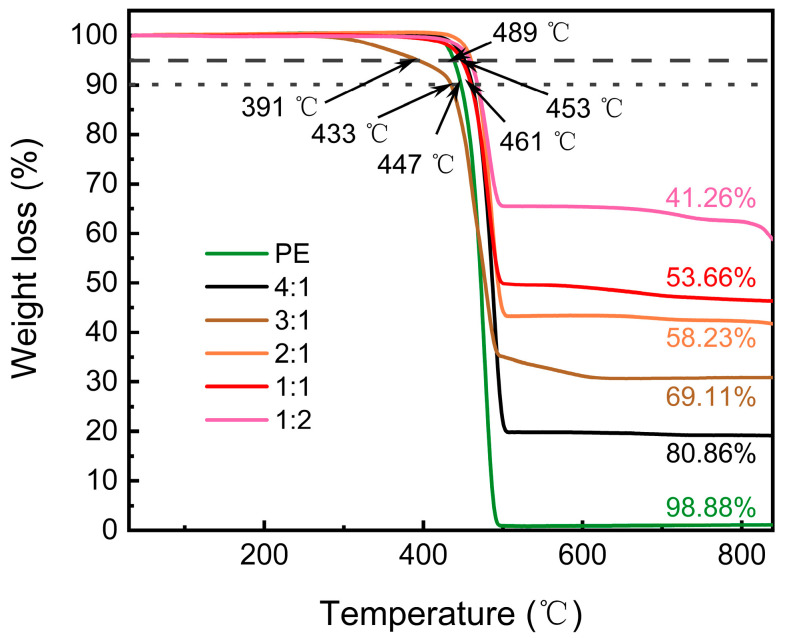
TGA diagram of different mass ratios between PE and 8%CNTs@Nd_0.15_-BaM.

**Figure 7 materials-17-03433-f007:**
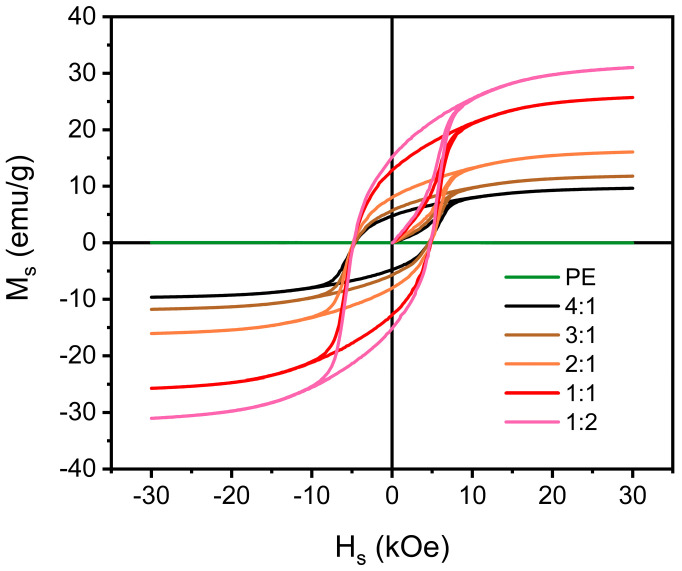
Hysteresis loop of different mass ratios between PE and 8%CNTs@Nd_0.15_-BaM.

**Figure 8 materials-17-03433-f008:**
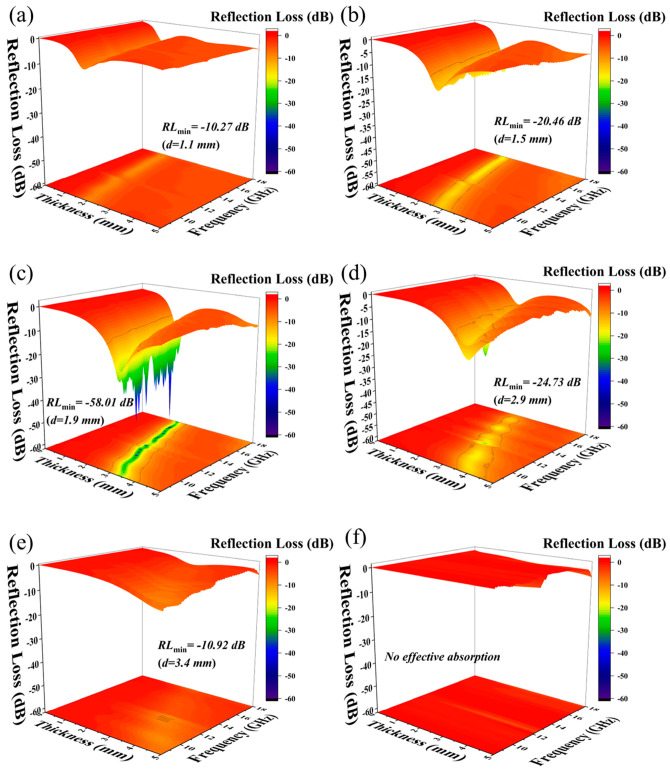
Reflection loss in the frequency of 8–18 GHz for the different mass ratios between PE and CNTs@Nd_0.15_-BaM ((**a**)—1:2; (**b**)—1:1; (**c**)—2:1; (**d**)—3:1; (**e**)—4:1; (**f**)—PE).

**Figure 9 materials-17-03433-f009:**
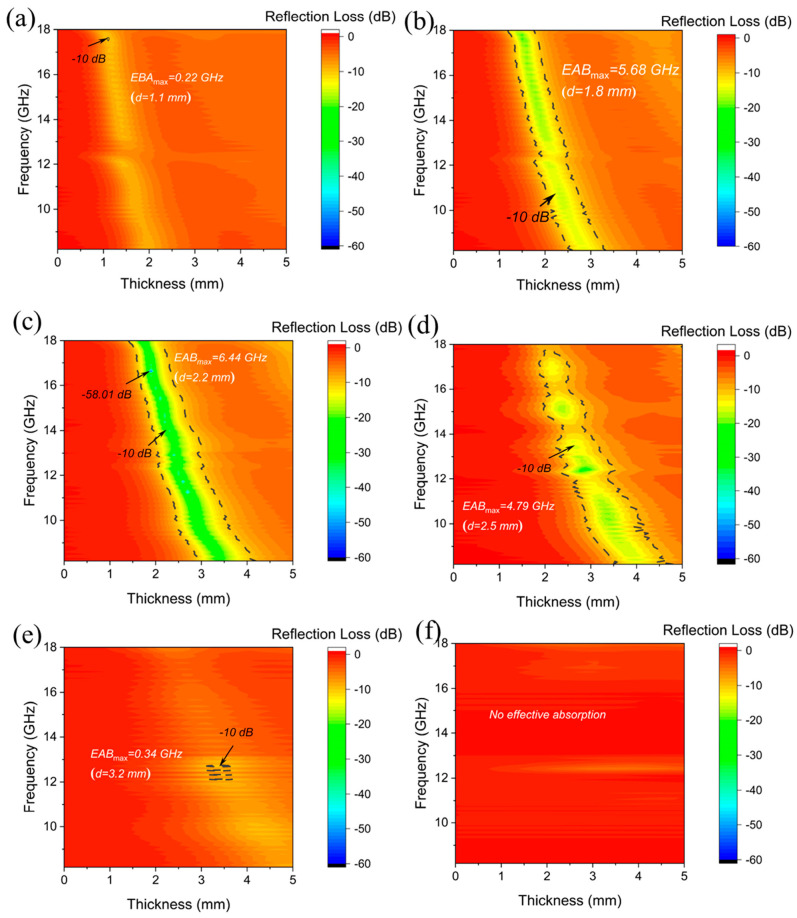
Contour map of absolute values for different mass ratios between PE and 8%CNTs@Nd_0.15_-BaM ((**a**)—1:2; (**b**)—1:1; (**c**)—2:1; (**d**)—3:1; (**e**)—4:1; (**f**)—PE).

**Figure 10 materials-17-03433-f010:**
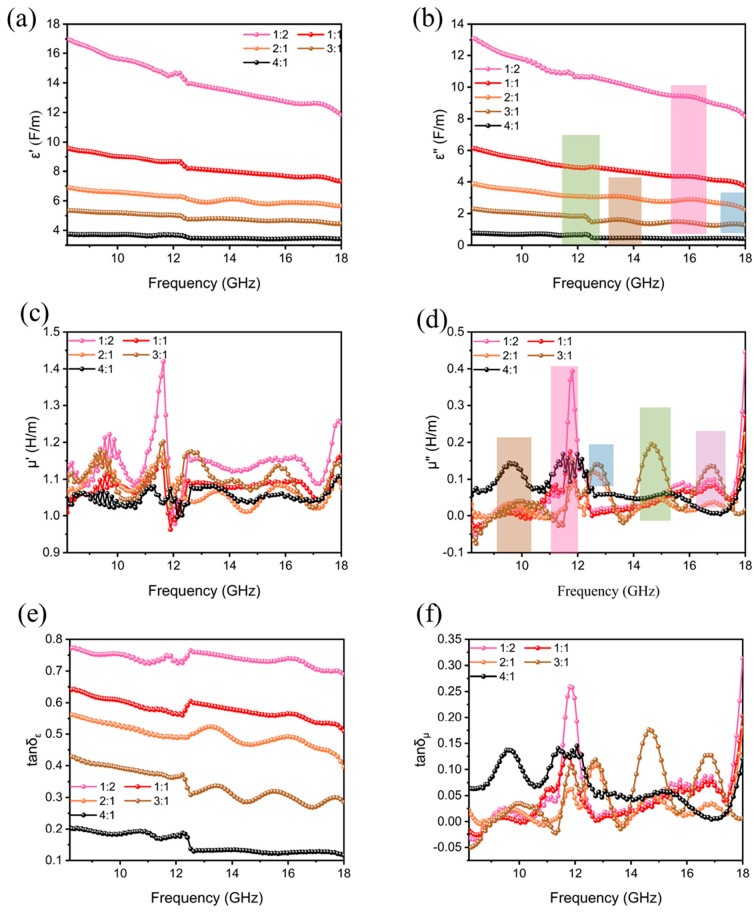
Complex dielectric constant (a-real part, b-imaginary part, c-tangent) and permeability (d-real part, e-imaginary part, c-tangent) of different mass ratios between PE and 8%CNTs@Nd_0.15_-BaM ((**a**)—1:2; (**b**)—1:1; (**c**)—2:1; (**d**)—3:1; (**e**)—4:1; (**f**)—PE).

**Figure 11 materials-17-03433-f011:**
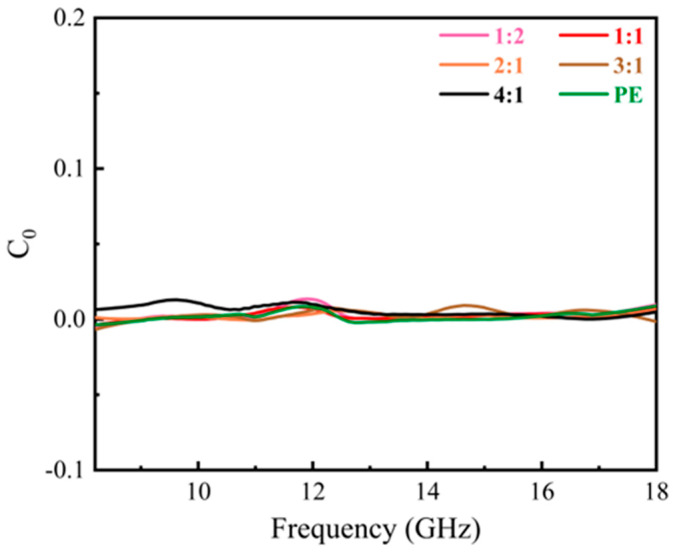
C_0_ of different mass ratios between PE and CNTs@Nd_0.15_-BaM.

**Figure 12 materials-17-03433-f012:**
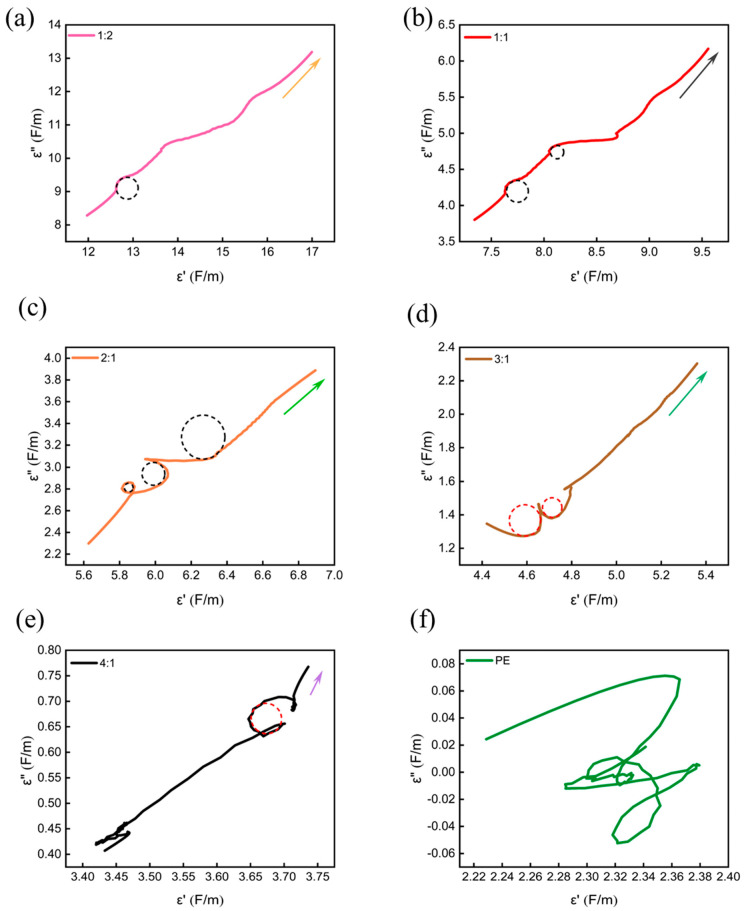
Cole–Cole semicircle curves for different mass ratios between PE and CNTs@Nd_0.15_-BaM ((**a**)—1:2; (**b**)—1:1; (**c**)—2:1; (**d**)—3:1; (**e**)—4:1; (**f**)—PE).

**Figure 13 materials-17-03433-f013:**
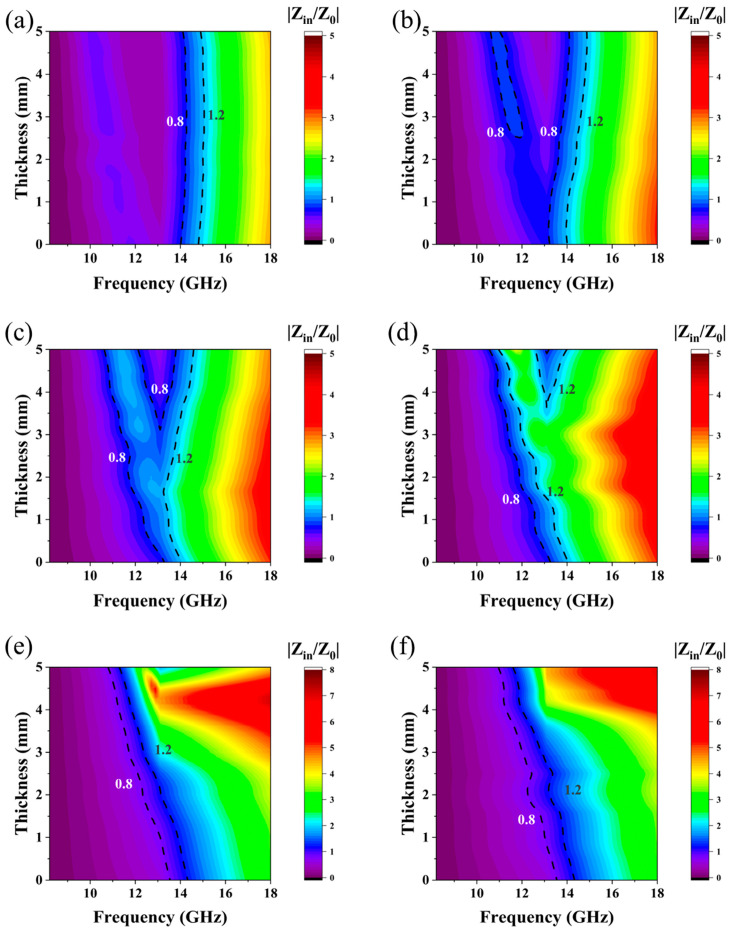
Normalized input impendence (Z_in_/Z_0_) of 1:2 (**a**), 1:1 (**b**), 2:1 (**c**), 3:1 (**d**), 4:1 (**e**), and PE (**f**).

**Table 1 materials-17-03433-t001:** Summary of the microwave absorption performance of BaM doped with different ions and composited of different carbon materials.

Materials	Methods	Shape	RL_min_ (dB)	The Frequency of RL_min_ (GHz)	EAB (GHz)	Thickness (mm)	Ref.
Al-BaM	Self-propagating combustion	powder	−34.76	14.57	/	/	[13]
Mn, Cu, Ti-BaM	Conventional ceramic technique	powder	−51.78	18.78	2.79 (<−20 dB)	1.8	[14]
Co-BaM	Solid-state reaction	powder	−32.1	11.2	5	2	[15]
Bi-BaM	Sol–gel	powder	−43.60	/	6.3	2.4	[16]
Sr, Cu, Zr-BaM	Sol–gel	powder	−15.20	11.1	/	/	[17]
Sr-BaM	Co-axial electrospinning	coating	−12.69	11.68	/	/	[18]
La-BaM	Electrospinning Heat treatment	powder	−23.03	2	12.6	2	[19]
Gd-BaM	Sol–gel	coating	−27.00	/	10	1.92	[20]
Nd-BaM	Sol–gel	powder	−17.91	12.56	/	3	[21]
MWCNTs/BaM	Sol–gel	powder	−56.47	4.8	6.2	1.35	[22]
C/BaM	/	powder	−30	9.2	0.6	3	[23]
C/BaM	The hydrothermal carbonization and subsequent calcinations	powder	−73.42	/	/	1.40	[24]
C/BaM	Surface carbonized layers	powder	−35	15.5	2.0	5.5 mm	[25]
C, SiC/BaM	Hydrothermal method	powder	/	/	8.8	2.8 mm	[26]
GO/Nd-BaM	Sol–gel	powder	−82.07	12.65	6.08	2 mm	[21]

## Data Availability

The raw data supporting the conclusions of this article will be made available by the authors on request.

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
