# Peer review of "Investigating Enhanced Microwave Absorption of CNTs@Nd0.15-BaM/PE Plate via Low-Temperature Sintering and High-Energy Ball Milling"

_materials, 2024, doi:10.3390/ma17143433_

Round 1

Reviewer 1 Report

Comments and Suggestions for Authors

The document is attached

Comments on the Quality of English Language

Author Response

The authors developed a PE/CNTs@Nd0.15 BaM composite sheets using sol gel, ball milling, and hot-pressing processes. Optimal microwave absorption was achieved at a mass ratio of 2:1 between PE and CNTs@Nd0.15 BaM. The sample showed 40.21 dB absorption at 13.86 GHz with 6.44 GHz bandwidth at 2.2mm thickness and 58.01 dB absorption at 1 6.43 GHz with 4.26 GHz bandwidth at 1 9 mm thickness.

COMMENT

The article presents interesting results, but some improvements are necessary for publication:

  1. Table 1 lists the value of RLmin without specifying what this value represents, and the units of measurement are inconsistently written with and without parentheses. I suggest standardizing the notation to make the table more readable.

Response: Thanks for your suggestion. Table 1 presents the RLmin, which represents the minimum reflection loss value. The RL value is calculated using Equation 1, and RLmin refers to the minimum value within the tested frequency range for that particular sample. Additionally, based on your comments, we have corrected the representation of units in the paper and highlighted the modified sections in yellow.

  1. Reference [21] in the table does not show any values, so I recommend removing it. Additionally, references like [28], which do not present relevant values for the paper, should also be removed from the table.

Response: Thanks for your suggestion. We have removed the references [21] and [28] from Table 1.

  1. If possible, specify the full names of raw materials along with their abbreviations.

Response: Thanks for your suggestion. We have clarified the full names and abbreviations of the raw materials in the article. Specifically, we have noted the full name and abbreviation of CNTs at line 44, BaM at line 51, GO at line 77, and PE at line 90. Furthermore, we have also labeled the chemical compounds Fe(NO3)3·9H2O, Ba(NO3)2, Nd(NO3)3·6H2O, C6H8O7·H2O, and NH3·H2O at lines 104-106. The modified sections have been highlighted in yellow for clarity.

  1. The methodology for electromagnetic characterization lacks detail. What type of waveguide is used? Which mode is propagating within the guiding structure? What tools are used to evaluate ? and ? ? Since the paper's results primarily depend on these measurements, more emphasis should be placed on the evaluation methods and how the results were obtained.

Response: Thanks for your suggestion. We used rectangular waveguides, and the electromagnetic waves propagate along the axis of the tube in the waveguide. The X-band and P-band waveguides were used for propagation within the transmission line mode. The P9377B vector network analyzer we employed allowed for direct measurement of the material's permittivity (ε) and permeability (μ).

  1. Figures 7, 8, 9, 11, and 12 are illegible and should be enlarged. The color bars in Figures 7 and 8 should have the same scale to allow for comparison. The same applies to Figure 12.

Response: Thanks for your reminder. We have replaced all unclear graphics with high-resolution versions. Figures 8, 9, 10, 12, and 13 (originally labeled as Figures 7, 8, 9, 11, and 12) have undergone reformatting, and Figures 8, 9, and 13 (originally labeled as Figures 7, 8, and 12) have been given consistent scale replacements. Furthermore, we have sent all the figures in high-definition format to the editor.

  1. There is no mention of measurement errors. How many times were the measurements repeated for each sample? Were different samples made using the same methodology tested? The absence of error bars in the evaluation of ? and ? raises concerns that these considerations were not addressed.

Response: Thanks for your suggestion. We have repeated the testing process twice using the same method for each sample and selected the ε and μ values that exhibited the best microwave absorption performance. We have supplemented the data selection method in the text and highlighted the corrected section in yellow for clarity.

  1. I suggest omitting Figure 14, as it merely graphically represents a conclusion derived from measurements that, in my view, are currently incomplete.

Response: We sincerely appreciate this valuable suggestion. We have removed Figure 14 and its corresponding analysis content from the document.

Thanks again.

Reviewer 2 Report

Comments and Suggestions for Authors

The present article deals with a topical issue. It concerns investigating the dielectric properties of special materials that could be used in various fields. In particular, one possible area in biomedical engineering is a shielding element against unwanted EM. After studying the article, I draw the following conclusions:

The article needs to be more systematic and illogically conceived. It is questionable whether it was subjected to correction by the given author team after writing. The results obtained may be valuable, but the article cannot be recommended for publication in this form. It needs to be more robust, particularly in terms of the absence of theoretical background /necessary equations and verbal description/, in the light of the comments. 

Lines 133 and 134 present the equations for the two crucial quantities/parameters. It's of utmost importance to remove the word EQUATION from both equations. The use of correct symbols to represent the quantities is a significant aspect that needs to be addressed. In Equation 2, the authors have used impedance, which is generally a complex number. However, the authors have not adhered to the established symbology for its designation. The correct use of parentheses is also essential in the expression of the equation. This is because it's not clear from the expression whether it's the absolute value of the argument or the symbol for the parentheses. Equations and their size should be adapted to the text. Subscripts should not be italicised.

In Figures 1-3, the names of the quantities are given verbally on both axes. This needs to be corrected so that their symbols are given on the axes and the verbal name in the text. In addition, the units of the quantities should be given in square brackets. This is a crucial step to ensure the accuracy and clarity of the figures.

In the title of Figure 5, the method should not be given together with the abbreviation. The method's name should be introduced in the text, together with the abbreviation. The abbreviation of the method only needs to be given in the figure.

All figures in the article need to be reduced in size, taking into account the proportions of the font of the text.

In Figure 6, the "magnetic field" is on the horizontal axis. It is not clear what quantity is meant. Is it a modulus calculated from spatial coordinates? The authors need to explain this clearly in the text or give the relationship by which the value was calculated. Otherwise, it is misleading. The same applies to the term "magnetization". What is the quantity in question? In EM field theory, the term magnetization covers a wide area. An explicit explanation is needed.

The quality of Figure 9 is deficient. Please increase its resolution. At the same time, the parameters of the complex dielectric constant are given on the vertical axes. In what units are these parameters? It is necessary to add them, for example, even if they are dimensionless quantities. Nowhere in the previous section are these parameters explained. The theoretical part needs to be completed where the reader understands what the symbols mean and from what equations they can be defined. This is an essential part that a good article must contain.

In Figure 11, it's necessary to complete the units of each quantity for better understanding. The nature of the graphs provided requires a more detailed description of the behaviour of the curves, particularly the F-curve. This will help the reader to fully comprehend the data and its implications.

Equation 4 should be presented in the theoretical part of the paper, not after the results have been obtained. This is because the paper should be systematically and logically structured, with the theoretical framework introduced before the results. This will help the reader to follow the logical progression of the research.

Fig. 12. Normalized input impendence: Is it the complex impedance modulus?

Fig. 13 needs to be scaled down considerably. As an overview figure, I recommend including it in the theoretical part of the paper after the introduction.

The explanatory value of Figure 14 is practically nil. For what reason did the authors include it here. Again, it should be included in the introduction of the article. 

Author Response

The present article deals with a topical issue. It concerns investigating the dielectric properties of special materials that could be used in various fields. In particular, one possible area in biomedical engineering is a shielding element against unwanted EM. After studying the article, I draw the following conclusions:

  1. The article needs to be more systematic and illogically conceived. It is questionable whether it was subjected to correction by the given author team after writing. The results obtained may be valuable, but the article cannot be recommended for publication in this form. It needs to be more robust, particularly in terms of the absence of theoretical background /necessary equations and verbal description/, in the light of the comments.

Response: Thanks for your valuable suggestions. We have tried our best to polish the language in the revised manuscript.

  1. Lines 133 and 134 present the equations for the two crucial quantities/parameters. It's of utmost importance to remove the word EQUATION from both equations. The use of correct symbols to represent the quantities is a significant aspect that needs to be addressed. In Equation 2, the authors have used impedance, which is generally a complex number. However, the authors have not adhered to the established symbology for its designation. The correct use of parentheses is also essential in the expression of the equation. This is because it's not clear from the expression whether it's the absolute value of the argument or the symbol for the parentheses. Equations and their size should be adapted to the text. Subscripts should not be italicised.

Response: Your valuable suggestion is much appreciated. We have removed the term "equation" from the context where it is inappropriate. The vertical bars "|  |" in Equation 2 represent the modulus of a complex number, and we have referenced this notation to reference [33] in the document. Furthermore, we have adjusted the size of the equations throughout the document to ensure they are appropriately scaled with respect to the surrounding text.

  1. In Figures 1-3, the names of the quantities are given verbally on both axes. This needs to be corrected so that their symbols are given on the axes and the verbal name in the text. In addition, the units of the quantities should be given in square brackets. This is a crucial step to ensure the accuracy and clarity of the figures.

Response: Thanks for your valuable suggestions, which have been very informative for us. We have supplemented the meanings of the x-axis and y-axis coordinates for Figures 2-4 (originally labeled as Figures 1-3) in the main text, and the supplementary sections have been highlighted in yellow.

  1. In the title of Figure 5, the method should not be given together with the abbreviation. The method's name should be introduced in the text, together with the abbreviation. The abbreviation of the method only needs to be given in the figure.

Response: Thanks for your suggestion. We have removed the full name of TGA from the title of Figure 5 and introduced the full name of TGA in the main text. The corrected section has been highlighted in yellow.

  1. All figures in the article need to be reduced in size, taking into account the proportions of the font of the text.

Response: Thanks for your suggestion. We have adjusted the size of the images and resized the text within the images to ensure that the text size is consistent with the font size of the main text.

  1. In Figure 6, the "magnetic field" is on the horizontal axis. It is not clear what quantity is meant. Is it a modulus calculated from spatial coordinates? The authors need to explain this clearly in the text or give the relationship by which the value was calculated. Otherwise, it is misleading. The same applies to the term "magnetization". What is the quantity in question? In EM field theory, the term magnetization covers a wide area. An explicit explanation is needed.

Response: Thanks for your suggestion. We have utilized the Model-9 Comprehensive Physical Property Measurement System produced by Quantum Design Inc. (USA) to measure the magnetic hysteresis loops of the samples, thereby obtaining the magnetic parameters of the materials. The coordinates on the X-axis and Y-axis are the measured quantities, and we have modified the figures according to the reference document 21.

  1. The quality of Figure 9 is deficient. Please increase its resolution. At the same time, the parameters of the complex dielectric constant are given on the vertical axes. In what units are these parameters? It is necessary to add them, for example, even if they are dimensionless quantities. Nowhere in the previous section are these parameters explained. The theoretical part needs to be completed where the reader understands what the symbols mean and from what equations they can be defined. This is an essential part that a good article must contain.

Response: We sincerely appreciate this valuable suggestion. We have replaced all unclear graphics with high-resolution versions and clearly labeled the units for the complex permittivity (F/m), complex permeability (H/m), permittivity tangent (dimensionless), and permeability tangent (dimensionless) in the figures. We have also supplemented this information in the main text. All modifications have been highlighted in yellow within the text. Additionally, we have sent all figures in high-definition format to the editor.

  1. In Figure 11, it's necessary to complete the units of each quantity for better understanding. The nature of the graphs provided requires a more detailed description of the behavior of the curves, particularly the F-curve. This will help the reader to fully comprehend the data and its implications.

Response: Thank you for your valuable suggestions. We have refined the units for ε' and ε'' in Figure 12 (originally labeled as Figure 11) and conducted a more detailed analysis of the f-curve. These modifications have been highlighted in yellow within the text.

  1. Equation 4 should be presented in the theoretical part of the paper, not after the results have been obtained. This is because the paper should be systematically and logically structured, with the theoretical framework introduced before the results. This will help the reader to follow the logical progression of the research.

Response: Thanks for your valuable advice. We have adjusted the position of some equations and consolidated them in Section 2.3 to enhance the rigor and logical flow of the article. The revised sections have been highlighted in yellow.

  1. 12. Normalized input impendence: Is it the complex impedance modulus?

Response: Thanks for your suggestions. As you mentioned, the normalized input impedance refers to the modulus of the complex impedance.

  1. 13 needs to be scaled down considerably. As an overview figure, I recommend including it in the theoretical part of the paper after the introduction.

Response: Thanks for your suggestion. We have resized Figure 13 to a smaller version and repositioned it within the Introduction section.

  1. The explanatory value of Figure 14 is practically nil. For what reason did the authors include it here. Again, it should be included in the introduction of the article.

Response: Thanks for your suggestion. We have removed Figure 14 and its corresponding analysis content entirely from the document.

Thanks again.

Reviewer 3 Report

Comments and Suggestions for Authors

I have the following suggestions to improve the quality of the paper so that it can be further considered for publication. 

1) The author should focus on the potential applications of the synthesized material. And write a short comparison of the previous works on this topic. And how their work stands among the other related works. This will highlight the novelty and significance of their work. 

2) The X and Y labels of the figures are too big. Reduce the font size and make it proportional to the other text size in the figure. 

3) Equations are also written in large font size. 

4) The author needs to pay attention to the presentation and the flow of information throughout the paper. 

5) Figure 7,8,9 seems like a screenshot of the image. I suggest enhancing the quality of the figures. 

6) Can the fabrication of this material be done with the sol-gel process and dip-coating method? instead of a ball mining process. 

Comments on the Quality of English Language

none. 

Author Response

I have the following suggestions to improve the quality of the paper so that it can be further

considered for publication. 

  1. The author should focus on the potential applications of the synthesized material. And write a short comparison of the previous works on this topic. And how their work stands among the other related works. This will highlight the novelty and significance of their work.

Response: Thanks for your comment.

  1. The X and Y labels of the figures are too big. Reduce the font size and make it proportional to the other text size in the figure.

Response: Thanks for your suggestion. We have adjusted the font size in the images to match the font size in the main text for consistency.

  1. Equations are also written in large font size.

Response: Thanks for your suggestion. We have adjusted the font size of the equations throughout the document to match the font size of the main text and have highlighted the changes in yellow.

  1. The author needs to pay attention to the presentation and the flow of information throughout the paper.

Response: Thanks for your valuable comments. We have refined the overall structure of the article. For instance, we have rearranged some equations and consolidated them in Section 2.3 to enhance the rigor and logical flow of the article. All modifications have been highlighted in yellow within the text.

  1. Figure 7,8,9 seems like a screenshot of the image. I suggest enhancing the quality of the figures.

Response: Thanks for your valuable advice. We have replaced all unclear figures with high resolution figures. Also, we send all the figures to the editor in high-definition format.

  1. Can the fabrication of this material be done with the sol-gel process and dip-coating method? instead of a ball mining process.

Response: Thanks for your suggestion. In the doping stage, we used the sol-gel method. For the carbon material composite phase, ball milling is likely necessary. Directly adding the sol-gel into it may affect the doping. This is a good suggestion, and we will try it later.

Thanks again.

Reviewer 4 Report

Comments and Suggestions for Authors

 In this work, the authors synthesized a composite plate containing neodymium (Nd) doped M-type barium ferrite, carbon nano tubes, and polyethylene which could be used as an improved microwave absorber in the frequency range of 8.2 to 18 GHz.  The authors also carried out martial characterization using XRD, FTIR, TG, Raman, and SEM. The authors experimentally observed that the composite material showed a much-improved adsorbing characteristic. This work is of importance to the electromagnetic shielding research where development of an improved material is very important.

However, several issues have to be resolved before this paper can be published.

1.       Please use the full form of the XRD, FTIR,and TG in the abstract before using the abbreviations

2.       Why the authors choose the and 8.2 GHz to 18 GHz? What is the significant of this band?

3.       How the electromagnetic test structures were developed? What type of connectors/antennas were used? An image of the test structure would be helpful.

4.       850° C sintering was carried out in air or in Nitrogen?

5.       On the page 6, line 172, the authors described. “These findings suggest that during the preparation process, 172 the BaM powder is bonded together by PE, forming blocks”. Could please discuss what type of bond it was?

6.       In Fig. 3, why the defect is less for the ratio of 1:2?

7.       For Fig. 4, a table showing the mass ratio would be helpful.

8.       Please increase the fonts for Fig. 7 and Fig. 8.

Comments on the Quality of English Language

Minor editing is required. 

Author Response

In this work, the authors synthesized a composite plate containing neodymium (Nd) doped M-type barium ferrite, carbon nano tubes, and polyethylene which could be used as an improved microwave absorber in the frequency range of 8.2 to 18 GHz.  The authors also carried out martial characterization using XRD, FTIR, TG, Raman, and SEM. The authors experimentally observed that the composite material showed a much-improved adsorbing characteristic. This work is of importance to the electromagnetic shielding research where development of an improved material is very important.

However, several issues have to be resolved before this paper can be published.

  1. Please use the full form of the XRD, FTIR, and TG in the abstract before using the abbreviations.

Response: Thanks for your suggestion. We have supplemented the full forms of CNT (carbon nanotubes), XRD (X-Ray diffraction), FTIR (Fourier Transform Infrared Spectrophotometer), TGA (Thermo Gravimetric Analyzer), and SEM (Scanning Electron Microscope) in the abstract, and the modified sections have been highlighted in yellow.

  1. Why the authors choose the and 8.2 GHz to 18 GHz? What is the significant of this band?

Response: Thanks for your suggestion. The frequency range of 8.2-12.5 GHz is referred to as the X-band, while 12.5-18 GHz is known as the P-band. The P-band was the earliest band used in radar applications, while the X-band corresponds to electromagnetic waves with a wavelength of 3 cm. These two bands are currently the most commonly used in radar systems.

  1. How the electromagnetic test structures were developed? What type of connectors/antennas were used? An image of the test structure would be helpful.

Response: Thanks for your suggestion. We have conducted electromagnetic testing on the samples using the waveguide method of the P9377B Vector Network Analyzer. The diagram of the test setup is shown below.

  1. 850°C sintering was carried out in air or in Nitrogen?

Response: We were inspired by your comment and realized that we had overlooked some important details. The sintering process at 850°C was conducted in air.

  1. On the page 6, line 172, the authors described. “These findings suggest that during the preparation process, 172 the BaM powder is bonded together by PE, forming blocks”. Could please discuss what type of bond it was?

Response: Thanks for your reminder. We believe that PE and BaM powder should be physically bonded because the temperature of our hot pressing is not high.

  1. In Fig. 3, why the defect is less for the ratio of 1:2?

Response: Thanks for your suggestion. Based on our Raman testing and referencing other literature, we have reached this conclusion, though further evidence may be needed.

  1. For Fig. 4, a table showing the mass ratio would be helpful.

Response: We sincerely appreciate this valuable suggestion. We have made revisions in the paper.

  1. Please increase the fonts for Fig. 7 and Fig. 8.

Response: Your valuable suggestion is much appreciated. We have increased the font size in Figures 7 and 8.

Thanks again.

Round 2

Reviewer 1 Report

Comments and Suggestions for Authors

The paper has been improved in terms of structure and clarity, and all questions raised have been addressed satisfactorily. It is now ready for publication.

Author Response

Thank you for recognizing our work, thank you.

Reviewer 2 Report

Comments and Suggestions for Authors

The authors have submitted a revised version of the article for review. In light of the comments, they have responded and incorporated the requirements in question into the article. After reviewing the article, I found that the level of professionalism is higher; therefore, the article is more engaging for the reader. Despite the authors' efforts, there are still some minor /formal/ shortcomings in the article that need to be corrected, namely, in equation 2 and line 148, the correct designation for the impedance Z should be used, as it is a complex number. In electrical engineering, the symbol of a particular character /dot from a given character set/ placed above the symbol Z is used to denote it /and thus to distinguish it from an actual number/. This is very important because it changes the nature of the quantity in question. In a lossless environment, a given impedance is indeed an actual number. However, in a lossy environment, it is already a complex quantity whose angle may or may not be constant. I ask the authors for this correction. Other comments are related to the graphs in Figures 1-4. I ask the authors to standardize the font sizes and types when presenting quantities on horizontal and vertical axes. Also, the spacing between the names of the quantities and their units needs to be checked, as they are missing in some places. In Figure 2, the DEGREE symbol should be replaced by a symbol to indicate degrees.

After incorporating the comments, I recommend the paper for publication.

Author Response

The authors have submitted a revised version of the article for review. In light of the comments, they have responded and incorporated the requirements in question into the article. After reviewing the article, I found that the level of professionalism is higher; therefore, the article is more engaging for the reader. Despite the authors' efforts, there are still some minor formal shortcomings in the article that need to be corrected:

  1. Namely, in equation 2 and line 148, the correct designation for the impedance Z should be used, as it is a complex number. In electrical engineering, the symbol of a particular character dot from a given character set placed above the symbol Z is used to denote it /and thus to distinguish it from an actual number/. This is very important because it changes the nature of the quantity in question. In a lossless environment, a given impedance is indeed an actual number. However, in a lossy environment, it is already a complex quantity whose angle may or may not be constant. I ask the authors for this correction.

Response: Thanks for your suggestion. We have added a special character above the complex number  to indicate it as a complex number. The corrected content is highlighted in green.

  1. Other comments are related to the graphs in Figures 1-4. I ask the authors to standardize the font sizes and types when presenting quantities on horizontal and vertical axes. Also, the spacing between the names of the quantities and their units needs to be checked, as they are missing in some places. In Figure 2, the DEGREE symbol should be replaced by a symbol to indicate degrees.

Response: Your valuable suggestion is much appreciated. We have adjusted the formatting of Figures 1-4, ensuring there are spaces between the numbers and their units. The unit "Degree" on the horizontal axis of Figure 2 has been replaced with "°". Additionally, we have adjusted the font size and formatting for the horizontal and vertical axes as well as within the charts to make them more standardized.

Reviewer 4 Report

Comments and Suggestions for Authors

Thanks for answering all the questions. 

Comments on the Quality of English Language

Minor editing is required. 

Author Response

Thank you for your recognition of our work, we have made slight changes to the interpretation of the diagram as well as the English expression, so that the reader can read it more comfortably. Thanks again.